# The HROC-Xenobank—A High Quality Assured PDX Biobank of >100 Individual Colorectal Cancer Models

**DOI:** 10.3390/cancers13235882

**Published:** 2021-11-23

**Authors:** Stephanie Matschos, Florian Bürtin, Said Kdimati, Mandy Radefeldt, Susann Krake, Friedrich Prall, Nadja Engel, Mathias Krohn, Bianca Micheel, Michael Kreutzer, Christina Susanne Mullins, Michael Linnebacher

**Affiliations:** 1Clinic of General Surgery, Molecular Oncology and Immunotherapy, University Medicine Rostock, Schillingallee 69, 18057 Rostock, Germany; stephanie.matschos@med.uni-rostock.de (S.M.); florian.buertin@med.uni-rostock.de (F.B.); said.kdimati@med.uni-rostock.de (S.K.); mathias.krohn@med.uni-rostock.de (M.K.); b.micheel@keh-berlin.de (B.M.); christina.mullins@med.uni-rostock.de (C.S.M.); 2CENTOGENE GmbH, 18055 Rostock, Germany; mandy.radefeldt@centogene.com (M.R.); susann.krake@centogene.com (S.K.); 3Institute of Pathology, University Medicine Rostock, Strempelstraße 10, 18057 Rostock, Germany; friedrich.prall@med.uni-rostock.de; 4Department of Oral, Maxillofacial and Plastic Surgery, University Medicine Rostock, Schillingallee 69, 18057 Rostock, Germany; nadja.engel@med.uni-rostock.de; 5Medical Research Center, University Medicine Rostock, Schillingallee 69, 18057 Rostock, Germany; michael.kreutzer@med.uni-rostock.de

**Keywords:** PDX model, CRC, mutation analysis, histological examination

## Abstract

**Simple Summary:**

Considering recent research, it was established that the best experimental models to conserve biological features of human tumors and to predict individual clinical treatment success are patient-derived xenografts (PDX). Their recognized and growing importance for translational research, especially for late-stage preclinical testing of novel therapeutics, necessitates a high number of well-defined PDX models from individual patients’ tumors. The starting platform for the Hansestadt Rostock colorectal cancer (HROC)-Xenobank was the assortment of colorectal tumor and normal tissue samples from patients stored in our university biobank.

**Abstract:**

Based on our research group’s large biobank of colorectal cancers (CRC), we here describe the ongoing activity of establishing a high quality assured PDX biobank for more than 100 individual CRC cases. This includes sufficient numbers of vitally frozen (*n >* 30 aliquots) and snap frozen (*n >* 5) backups, “ready to use”. Additionally, PDX tumor pieces were paraffin embedded. At the current time, we have completed 125 cases. This resource allows histopathological examinations, molecular characterizations, and gene expression analysis. Due to its size, different issues of interest can be addressed. Most importantly, the application of low-passage, cryopreserved, and well-characterized PDX for in vivo studies guarantees the reliability of results due to the largely preserved tumor microenvironment. All cases described were molecularly subtyped and genetic identity, in comparison to the original tumor tissue, was confirmed by fingerprint analysis. The latter excludes ambiguity errors between the PDX and the original patient tumor. A cancer hot spot mutation analysis was performed for *n =* 113 of the 125 cases entities. All relevant CRC molecular subtypes identified so far are represented in the Hansestadt Rostock CRC (HROC)-Xenobank. Notably, all models are available for cooperative research approaches.

## 1. Introduction

Despite early diagnostic options and improved treatment, colorectal cancer (CRC) is still one of the leading causes of cancer-related deaths worldwide [1]. In particular, the fact that some patients do not respond even to targeted therapies underlines the necessity of further patient-derived models to promote the development of personalized treatments [1,2].

Currently the best model to reflect the characteristics of the original tumor is the patient-derived xenograft (PDX) model because of its conservation of the original tumor’s biological features, including microarchitecture, pathomorphology, and genetic alterations [2,3]. Tentler et al. have stated that CRC PDX tumors retain the intratumoral clonal heterogeneity, chromosomal instability, and histology of the parent tumor for up to 14 passages [2,4]. Furthermore, the possibility of precisely predicting individual clinical treatment success, especially for the late-stage preclinical testing of novel therapeutics, implies a clear exigency for more academically run PDX-biobanks, containing large numbers of individual tumors [2,3,5,6]. Inspired by this notion, we used our large collection of patient material, which included matching tumor and normal epithelial tissue, as a starting platform to establish a high number of individual PDX models. This resulted in a quality assured PDX biobank containing more than 100 individual CRC cases and encompassing all specific CRC molecular subtypes. Thus, this PDX biobank represents an ideal platform to study new agents for adjuvant therapy. As such, it is feasible to target specific molecular subtypes or alterations, in combination with investigations concerning different molecular pathways within the tumor cells as compared to the normal epithelial tissue. Such an approach has recently been described by Medico and colleagues. Here the authors identified tumor specific changes that consist of clinically actionable kinase targets for which approved drugs are already available [7]. Moreover, omics data from both the PDX model and the original patient tumor could, on the one hand, accelerate the entry of novel drugs into the clinic, and, on the other hand, such paired data sets would facilitate the identification and validation of predictive biomarkers [2].

Finally, as has been described by us and other groups, the PDX-derived tissue is an ideal source for repetitive cell line establishment [8,9] and also patient-derived organoid (PDO) generation attempts [9,10]. This can significantly boost the overall success rate, from 10–13% for primary patient material derived cell lines [8,11] to about 30% for secondary, i.e., PDX-derived, cell lines [8]. Our vision is an integrated biobank collection, consisting of deeply characterized primary patient material, 2D cell lines, PDX, and PDO. With this vision we would like to support orchestrated research strategies, from more basic mechanistic approaches to translational drug development and tests to end-stage preclinical studies. Besides dogmatic animal welfare policies, the establishment and proper long-term maintenance of platforms such as the integrated Hansestadt Rostock CRC (HROC) biobank are essential for minimizing the overall number of animals involved in oncological in vivo studies.

## 2. Materials and Methods

Surgically resected tissues were collected from consenting patients at the UMR from 2006 to 2019. The study was approved by the ethics committee of the UMR (II HV 43/2004, A45/2007, A2018-0054, and A2019-0187) [8,12].

### 2.1. PDX Generation

Tumor engraftment was performed according to the guidelines of the local animal use and care committee, Landesamt für Landwirtschaft, Lebensmittelsicherheit und Fischerei Mecklenburg-Vorpommern with the permit numbers: LALLF M-V/TSD/7221.3-1.1-071/10; 7221.3-1-015/14; and 7221.3-2-020/17. The mice strains used were bred in the animal facility of the Rostock University Medical Center and maintained in specified, pathogen-free conditions, exposed to 12 h light/12 h darkness cycles. The mice received standard pellet food and water ad libitum.

Prior cryopreserved matching patient-derived tumor and normal tissue samples in our biobank of CRC patients served as the starting platform for all established PDX models. Detailed information on the patients’ tumors, as well as clinico-pathological information, is given in Appendix A.

Pieces of the patients’ tumors were implanted subcutaneously into the animals’ left and right flanks, under anesthesia (ketamine/xylazine, 90/6 mg/kg bw). Due to the engraftment rate of up to 80%, the preferred mouse strain for this first passage is NOD.Cg-Prkdc^scid^ Il2rg^tm1Wjl^/SzJ (NSG). Further passaging can be performed either with NSG or with NMRI-*Foxn1*^nu^ (NMRI nude mice). The PDX tumors were named with anonymized patient information followed by the abbreviation Tx, standing for the passage of the PDX tumor, and then followed by the abbreviation Mx, standing for consecutively numbered mice. All tumor engraftments were performed on 6–12 week-old mice, both male and female, weighing 18–30 g. Prior to xenografting, four vital tumor aliquots (3 × 3 × 3 mm) were soaked in 100 µL Matrigel (Corning, Kaiserslautern, Germany) for >10 min at 4 °C. After 30 days of antibiotic treatment (drinking water containing cotrimoxazole: dosage 8 mg trimethoprim and 40 mg sulfamethoxazole per kg BW), tumor growth was monitored weekly until tumor establishment and growth to a maximal diameter of 14.2 mm. When the maximum tumor volume of 1500 mm^3^ was reached or the mice became moribund, the tumors were explanted. The time of tumor growth until explantation was defined as tumor harvesting time.

After the explantation of the PDX tumors, the tumors were stored in Tissue Storage Solution (Miltenyi, Bergisch-Gladbach, Germany) until further processing. Snap frozen aliquots were made as soon as possible by immediately submerging tumor pieces in liquid nitrogen to ensure high quality, particularly for RNA molecules. Vital aliquots were made by transferring four tumor pieces of 3 × 3 × 3 mm in 1.5 mL freezer medium (fetal bovine serum with 10% DMSO) and cooling them down in CoolCell^®^ LX—freezing containers (CryoShop, München, Germany) by −1 °C per minute to −80 °C [13].

While processing the PDX tumors, the degree of necrosis was assessed and documented, allowing a classification into not necrotic, barely necrotic, intermediately necrotic, and highly necrotic.

The process describing our PDX biobank establishment approach was recently published [14].

### 2.2. Histopathology

For each PDX model, one representative cross section of a subcutaneous PDX tumor or half of the PDX tumor was fixed immediately upon explantation in formalin and embedded in paraffin by routine procedures. H&E-stained sections (4–5 μm) were analyzed in light-microscopic studies to assess the morphologic features of each individual PDX model [15]. A comparison with the respective original patient tumor was performed by a board-certified pathologist (FP).

### 2.3. Quality Control via Short Tandem Repeat (STR) Analysis

The fluorescence-labeled, PCR-amplified DNA fragments of D5S818, D7S820, D16S539, D13S317, vWA, TPOX, THO1, CSF1PO, and Amelogenin were injected along with an appropriately sized standard GeneScan™ LIZ500 (appliedbiosystems Thermo Fisher Scientific, Waltham, MA, USA) into the capillary for electrophoresis size separation, using ABI instrumentation. Size-separated PCR fragments were detected by reading their fluorescence intensity at different emission wavelengths and were recorded as FSA after their migration through the capillary from cathode to anode, in which smaller fragments migrate faster than larger fragments [16]. The application of primer pairs labeled with three different fluorescence dyes—FAM (blue), HEX (green), and TAMRA (red)— enabled the fragment size determination of all markers mentioned (primers listed in detail in Table 1) in a single analysis.

### 2.4. Molecular Classification Analyses

The microsatellite instability (MSI) and methylation status of CpG islands [17,18,19] were determined for all cases included in this study. The classification was MSI-H if two or more microsatellite markers of either the Bethesda panel or the “six mononucleotide repeat” panel, consisting of BAT25, BAT26, CAT25, NR21, NR24, and NR27, showed band shifts [8,17]. Classification concerning the CpG island methylator phenotype (CIMP) was carried out as follows: if the analysis was performed according to Ogino et al., the subtype was divided into CIMP-H, non MSI, when ≥4 loci, and CIMP-L, non MSI, if 1–3 CIMP loci out of 5 loci analyzed were methylated [17,18]. When analyzed according to Weisenberger et al., ≥3 methylated CIMP Loci out of 5 loci analyzed defined CIMP-H, non MSI; no further distinction of CIMP-L, non MSI took place [19].

### 2.5. Next Generation Sequencing (NGS) Analyses

In total, 121 datasets were either generated by Centogene (Rostock, Germany) or extracted from a previous dataset [20]. This dataset consisted of Whole Exom Sequencing (WES) analyses of 20 PDX cases and 12 primary tumors. The remaining analyses were performed using a Solid Tumor Panel from Centogene consisting of 105 fully sequenced genes, plus mutational hot spots from an additional 146 genes. Library preparation was performed with the Twist Library Preparation Enzymatic Fragmentation Kit (Twist Bioscience, San Francisco, CA, USA). Exome enrichment was carried out, using either the TWIST Human Core Exome Plus probes (covering 36.5 Mb of the human coding exome) or custom designed probes, in the case of the Solid Tumor panel. Sequencing was performed using the NextSeq500 (Solid Tumor panel) or the HiSeq4000 and NovaSeq (WES) systems (Illumina, Inc., San Diego, CA, USA) to produce 2 × 150 bp reads. Raw sequencing reads were converted to standard fastq format using bcl2fastq software 2.17.1.14 (Illumina, Inc., San Diego, CA, USA). The short-reads were aligned to the GRCh37(hg19) build of the human reference genome using Bowtie version 2.4.2 [21]. The alignments were sorted (samtools v. 1.11) [22] and de-duplicated (PicardTools v. 2.23.8) [23]. Variant calling was performed with Strelka Somatic pipeline (v. 2.9.2) [24]. The variant table was filtered with vcftools v. 0.1.16 [25] and annotated with snpEff [26]. The filters applied were set to protein-coding mutations, filter “Pass”, allele frequency > 5%, quality > 50, and at least 20 reads for the tumor.

For one PDX case, tissues obtained from two different mice were analyzed, and for a second PDX case, tissues obtained from two different passages were analyzed.

Concerning the NGS data for this study we focused on mutations being pathogenic or likely pathogenic, but also included mutations of uncertain significance. Excluded were all benign mutations, as well as mutations classified as risk factor and influencing drug response. Moreover, only the mutations from the raw data which passed the following quality criteria, including the filter “Pass”, ≠coding synonym, a quality ≥30, and a variant allele frequency of at least 15, were listed.

### 2.6. Statistical Analyses

Statistical analyses were performed using either the statistical program prism 8 or IBM SPSS Statistics. Heatmap and mutation frequency analyses were performed in Prism. In SPSS, a nonparametric bivariate correlation analysis according to Kendall-Tau and Spearman’s rank correlation coefficient and Fisher’s exact test were performed. The cluster analysis was performed with Origin Pro 2017G (parameters: cluster method = Furthest Neighbor; distance type = Euclidean).

## 3. Results

In order to achieve maximal quality and traceability, the data from the CRC PDX cases included in the end, were collected and were mostly presented according to the PDX-Minimal Information standard (PDX-MI) recommended by Meehan et al. [27]. The PDX-MI suggests four modules reflecting the process of generating and validating a PDX model: (1) clinical data, (2) model creation data, (3) model quality assurance data, and (4) model study and associated metadata. Detailed information on our analyzed PDX cases was arranged accordingly and listed in Appendix A.

### 3.1. Patient, Clinical and Molecular Tumor Data

In total, 261 CRC patients were included in this study in the time span of October 2006 to May 2019. From these cases, 167 individual PDX models could be generated (64.0%). The present study focuses on 125 of these cases, which have been selected according to the following criteria: (I) enduring growth in immunodeficient mice and (II) storing sufficient quantities of PDX tissues with (III) adequate quality. The latter criteria in particular, led to the exclusion of 20 cases (12.0%) due to very high proportions of necrotic areas reproducibly observed in the harvested PDX tissues. Analyses of 22 PDX are not yet finalized, and thus these were consequently excluded from the present study.

To anonymize the patient information, each case was assigned an alias consisting of: HRO for Hansestadt Rostock, C for colon cancer, and a consecutive number. Metastases included were given the identifier Met as an abbreviation of metastasis. This was added directly after the HROC number. In case of multiple tumors, an additional tumor numeration was included.

All available patient information following surgical removal of the tumor, e.g., further treatments, disease recurrence, progression free and overall survival, were collected as described before [8] and updated in May 2020. These data are listed in Appendix A. The patient tumor samples consist of 100 primary adenocarcinomas, including one neuroendocrine tumor. Twenty-five samples are of metastatic origin, largely of the liver (80.0%). Metastases also manifested in the abdominal wall, brain, lung, peritoneum, and multivisceral (*n* = 1, each).

The gender distribution of the 125 patient cases included was 56.8% male and 43.2% female. The mean age was 69.7 years (ranging from 30 to 98). Tumor UICC staging was 11% stage I, 29% stage II, 27% stage III, and 33% stage IV. T stages were 30% T4, 59% T3, 10% T2, and 1% T1; M stages 1% M2 and 32% M1; 67% had no metastases identified (M0). Tumor grading (G) was 2% G1, 55% G2, and 43% G3.

Due to the fact that the integrated biobanking activities started in 2006 and are an ongoing process, the included cases cover the time period of 2006 to 2019. Thus, it was not possible to calculate the 5 year survival rate for all patients. Accordingly, setting a cutoff for the calculation of survival time was necessary. At the cutoff of May 2020, 54 patients were still alive, and 71 patients were dead. Three patients died perioperative, within 30 days after surgery (congruent with Clavien-Dindo classification). For the remaining 68 deceased patients, progression free survival averaged 13.0 months (ranging from 0 to 119) and the mean overall survival was 32.6 months (ranging from 1 to 133).

Each patient’s individual cancer history and therapy regiment was listed in detail if applicable, including type and duration of therapy, as well as applied chemotherapeutic agents (Appendix A). Because most of our in-house therapeutic studies with different agents compared to the standard of care are ongoing or will be published soon, the therapeutics which showed a reduction in the PDX tumor growth compared to the standard of care are simply listed in Appendix A.

Since an ideal CRC PDX collection should approximate the molecular heterogeneity of clinical cases, our 125 PDX were classified according to the following molecular subtypes [17]: chromosomal instable (CIN), sporadic microsatellite instable (spMSI), having the CpG island methylator phenotype (CIMP, sub classified into high level (CIMP-H) and low-level (CIMP-L)), and Lynch syndrome (LS) (Table 2).

The distribution of the molecular subtypes mostly corresponds to the general clinical distribution [28]. Only spMSI-H and LS are overrepresented, which is most likely attributable to the high engraftment rates of these molecular subtypes [29].

### 3.2. Biobanking of Established HROC PDX Models

Besides mirroring the clinical characteristics of patient cohorts, another major goal was to generate and biobank ample amounts of PDX tissue for subsequent analyses and future preclinical studies. In particular, *n* ≥ 30 vital PDX tissue backups (consisting of four small cubes of approximately 3 mm side-length, to allow for a total of at least 120 implantations), plus a minimum of *n =* 5 snap frozen samples, ideally suited for molecular analyses, were generated and stored in the gas phase above liquid nitrogen for each case. It is notable that the backups of all 125 cases were generated within less than 10 passages, and usually in less than 5 passages. This ensured closest achievable proximity to the tissue of origin. Moreover, representative cross-sections and halves of the PDX tumors were fixed in formalin and paraffin-embedded (in the following, this is termed FFPE tissue) for histopathological assessment.

The HROC Xenobank contains eight sets of primary tumor and metastases derived tissues from the very same patients; namely: HROC72 and HROC72Met1; HROC147 and HROC147Met1; HROC277, HROC277Met1 (synchronous), and HROC277Met2 (metachronous); HROC278 and HROC278Met1; HROC300 and HROC300Met1; HROC348 and HROC348Met1; HROC362 and HROC362Met1; and HROC405 and HROC405Met1. Additionally, three sets of two metastases from the very same patient are included: HROC103Met1 and Met2, HROC230Met1 and Met2, as well as HROC313Met1 and Met2. Furthermore, two sets of different primary tumors from the very same patients are included as well: HROC252Tu1, Tu2, and Tu3, plus HROC386Tu1 and Tu2.

The mean duration from implantation to harvest for all the included PDX models and overall passages was 105 days (range 38 to 287). No significant differences in duration until harvest were observed for the initial passages with 120 days (range 36 to 329) compared to 106 days (range 35 to 324) for the last performed passages of the included PDX models. Mice presenting health conditions leading to premature harvest were excluded from calculation of duration to harvest. These cases are indicated with >x days in Appendix A. Harvesting times were compared between the first and the last passage of each individual patient case, and increased or decreased growth is indicated with arrows in Appendix A column AG and column AH. A trend for shorter duration until harvest for the last passage was observed in the majority of cases: 70/125 (56%). No correlations with molecular subtype or features of the PDX became apparent.

Moreover, a direct comparison of the utilization of fresh vs. vitally frozen tissues for subsequent passaging was possible for 28 cases (Appendix A). In 23 of those cases (82.1%), a shorter time to harvest was observed when PDX tissues were passaged fresh (unpaired *t*-test *p* = 0.0003).

A correlation analysis revealed, beside the expected positive correlations between the UICC stage and patient’s progression free and overall survival, correlations between the PDX model features and the properties of the patient tumor (Appendix A). Further, 62.0% of the PDX established from male patients originated from patients in the age group 50–69 years old, whereas 53.7% of the tumors from female patients originated from patients > 70 years. Concerning only primary tumors, the distribution of models derived from male or female patients is balanced at 50.5% vs. 49.5%, but concerning metastases, more models derived from male patients are represented, with a division of 80.0% vs. 20.0%. Moreover, within the molecular subtypes, male patients’ tumors dominated within the CIN cases (70.8%). Cases of the subtype spMSI-H were predominantly from female patients (79.3%).

Further, the molecular subtype and the time to harvest of PDX tumors were correlated (correlation *p* < 0.001, exact fisher test *p* = 0.008). Moreover, the number of mice necessary to generate sufficient backups correlated with the molecular subtype (correlation *p* = 0.002, exact fisher test *p* = 0.048). We observed again a correlation between the localization of the tumor and the duration until harvest (correlation *p* = 0.002, exact fisher test *p* = 0.047). Tumors of the right colon generally needed less than 90 days. For the other localizations, the duration was 90–180 days until outgrowth. Concerning the molecular subtype, it can be pointed out that MSI-H tumors, both sporadic as well as Lynch-associated, for the most part grew out in less than 90 days. The remaining molecular subtypes took 90–180 days to reach the designated size. Ample backups could, for most PDX cases of the spMSI-H type, be generated using fewer than 3 mice, whereas the highest numbers of mice, frequently > 4.5 mice, were necessary for CIMP-L, non MSI-type tumors. Furthermore, it is worth mentioning that there is a positive correlation between the time to harvest of the last passage performed and patients’ overall survival (correlation *p* = 0.033, but not significant in exact fisher test).

### 3.3. Identity Testing

The genetic identity of all established PDX models in comparison to the original patient tissue was confirmed by fingerprint analysis as described before [30]. The PDX were found to be either genetically identical to or descended from the respective patient (data not shown) with two exceptions: no evaluable signals could be generated by fingerprint analysis for PDX HROC32 T3 M7 and HROC223 T2 M1. Allelic imbalances were regularly observed as well as small shifts in allele length for PDX of the MSI molecular subtype. These phenomena are both well-known. However, the possibility of comparing different patient tissues, as well as primary and secondary cell lines in many of the MSI-cases, still allowed valid identity verification.

### 3.4. Histological Examination

H&E-stained sections from FFPE blocks were used to assess morphologic features of each individual PDX model. Moreover, PDX tumors were compared to the respective original patient tumor by a board-certified pathologist (FP). Figure 1 depicts three selected cases: HROC172, HROC260, and HROC386Tu1. Details of the histological investigations are listed in Appendix A.

A comparison was not possible for 17 out of the 125 cases due to a lack of patient tumor FFPE material, thus allowing a direct comparison for a total of 108 cases. Concordance of patient and PDX tumor structure was found in 92 cases (85.2%), minor differences were noticed 11 times (10.2%) and marked differences occurred in 5 cases (4.6%). Here, a fingerprint analysis performed with gDNA, isolated from sections of the very same FFPE tissue blocks used for the histological examination, confirmed genetic identity with their respective patients of origin for three out of the five cases (HROC251, HROC370, and HROC447). For the cases HROC32 and HROC223, the pathologist suspected heavy contamination of the PDX tissues with murine or human lymphatic cells. Since the fingerprint analysis failed for these two PDX tissues, as mentioned above, species-specific PCR analyses were performed. The results allowed the conclusion that murine thymoma cells predominated in the PDX tissues. Of note, identity tests from HROC32 PDX tissue after the initial mouse passage, as well as from two PDX-derived cell lines generated from the same passage as the FFPE tissues, matched the patients’ identities.

A side-by-side comparison of PDX tumors derived from the same passage, but different animals, for six cases with different molecular subtypes (HROC92, HROC111Met1, HROC131, HROC169, HROC324, and HROC430) delivered exactly matching pathomorphological results (Appendix A). Thus, we would conclude that the conservation of the original tumor’s biological features such as microarchitecture and pathomorphology are an intrinsic feature of the individual model and we expect these to be stably maintained for several passages as shown before by Tentler et al. [2].

Finally, when PDX tumors were explanted, the degree of necrosis was assessed. Although this is not a very precise method, it allowed the classification of the PDX cases into four categories: not, barely, intermediately, and highly necrotic (Appendix A, column AJ). Here, we observed no significant correlations between the degree of necrosis and patient data. The MSI cases, both sporadic and LS, the PDX models which needed fewer mice for complete asservation, and the PDX cases with shorter duration until harvest were rarely highly necrotic: there were only 4/39 cases (10.3%) vs. 27/86 cases (31.4%) for the remaining molecular subtypes. Highly necrotic PDX tumors maintained this characteristic also in later passages. The paired PDX cases of primary and metastasis derived tumors from the very same patients (*n* = 8) always had very similar necrosis categories (Appendix A).

### 3.5. Mutation Analysis

Selected cases (*n =* 113) were analyzed using a Solid Tumor Panel NGS approach. The NGS data are presented in detail in Appendix A. For two cases (HROC277Met2 and HROC405), two individual PDX tumors were analyzed. Here, the same pathogenic or likely pathogenic mutations were observed. The PDX tumor with the lower passage for patient HROC405 presented additional mutations of uncertain significance. Furthermore, for four cases, NGS analyses were conducted with patient and PDX tumor tissue (HROC285, HROC404, HROC415Met1, and HROC419). Pathogenic or likely pathogenic mutations detected in the original patient tumors were also detected in the PDX tumors, with one exception: a mutation in RHOA was only found in the primary patient tumor of HROC419. However, the PDX tumors of HROC285, HROC404, and HROC415Met1 displayed additional pathogenic or likely pathogenic mutations. In case of HROC285, four additional mutations in the genes *ABCB4*, *KRAS*, *MSH2*, and *NF1* were observed, whereas HROC404 and HROC415Met1 displayed two additional mutations in *AXIN2* and *HRAS*, as well as in *KMT2D* and *KRAS*, respectively. Besides, the PDX tumors displayed more additional mutations of uncertain significance than the original tumors.

Next, an unsupervised cluster analysis including only the pathogenic or likely pathogenic mutations, was performed (Figure 2).

The clusters highlighted in orange contain mostly CIN cases (70.0 and 65.7%), whereas the cluster highlighted in yellow consists almost exclusively of sporadic MSI tumors (88%). However, the cluster highlighted in red could not be linked to a specific molecular subtype.

Furthermore, the number of pathogenic and likely pathogenic mutations detected per gene and case are illustrated in a heat map (Figure 3A). The mutation frequency of each gene can be found in Figure 3B. The most frequently mutated genes in the HROC-Xenobank are *APC* (50.4%), *KRAS* (39.8%), *TP53* (37.2%), *BRAF* (23.0%) and *PIK3CA* (17.7%).

## 4. Discussion

In summary, in this study we succeeded in establishing a CRC xenobank containing 125 individual PDX models with sufficient numbers of vital backups, snap frozen aliquots for molecular analysis, and FFPE material.

Mattar et al. described the challenges of creating a PDX biobank and proposed requirements for sample characterization and validation [31]. These included the collection of the above-mentioned sample types, namely, vital tissue, snap frozen samples, and FFPE specimens. In addition, they urged for genomic profiling and comparative histological reviewing. All of these recommendations were followed in our study. Comparative pathomorphological analysis and genetic identity testing confirmed the close proximity of the HROC-Xenobank models to the original patient tumors. Moreover, we could confirm previous findings that PDX models, in the majority of cases, maintain the original tumor’s biological features [2,3,32]. For each individual HROC PDX model we described in detail, the pathomorphological structures were reproduced from the original patient tumor. A side-by-side comparison of PDX tumors derived from the same passage but different animals selected randomly revealed exactly matching pathomorphological results. Thus, we concluded that the original tumor’s biological features, such as microarchitecture and pathomorphology, are intrinsic features of the individual tumor; thus, they are also conserved in the derived models and are most likely stable for several passages. It has been shown before that CRC PDX retain the histology as well as other features of the parental tumor, including intratumoral clonal heterogeneity and chromosomal instability, for up to 14 passages [2].

A surprising observation of the present study was the comparable duration until harvest for PDX tumors of the first and the last performed passage, at least with regard to mean values. Others have described a significantly accelerated growth rate with increasing passages [33]. One possible explanation for this discrepancy might come from the fact that we used different mouse strains. Because the engraftment efficacy for CRC ranges between ~60 and 70% in NMRI nude mice and up to ~80–90% in NSG mice [34], the preferred mouse strain for the first passage was NSG. Subsequent passaging was performed either with NSG or NMRI nude mice. The latter strain was preferred, since the risk of murine and/or human lymphoma development is reduced [35,36,37]. When considering this, we cannot formally exclude the possibility that this may have biased our results and this might explain why similar harvesting times between different passages were observed. Another factor affecting the time to harvest is the tissue condition at the time of engraftment, i.e., fresh vs. vitally frozen samples. For 23/28 (82.1%) cases, passaging with fresh material resulted in a significantly diminished duration until harvest.

Abdirahman et al. reported an engraftment success of 22/33 (67%) cases for their CRC PDX series, but because four cases turned out to be human lymphomas, their rate dropped to 18/33 (55%) [32]. Their lymphoma rate was surprisingly high (4/22; 18.2%). When comparing this with the 1.6% (2/125) rate of murine lymphoma in our study, and considering that Abdirahman and colleagues performed all passaging in NSG mice, it becomes clear why switching from NSG for initial engraftment to passaging in NMRI nude mice is preferable.

Further, the HROC PDX models precisely recapitulated the mutation profiles of the original patient tumors, thereby confirming previous data [32,38,39]. In particular, our study pointed out that pathogenic or likely pathogenic mutations detected in the original tumors were maintained in the PDX tumors. The current gold standard of NGS data analysis to compare are the data contained in the Cancer Genome Atlas Network (TCGA). In comparison to the TCGA results, the HROC-Xenobank mutational landscape differed in parts. The frequency for the most commonly mutated gene, *APC*, was 72.5% in the TCGA data set, compared to our frequency of only 50.4%. The *PIK3CA* gene had a frequency of 27.5% in TCGA and of 17.7% in the HROC-Xenobank. Frequencies for *BRAF* mutations were 11.6% (TCGA) and 23% (HROC-Xenobank). The most striking difference was noticed for *TP53* mutation frequency. Here 58.8% were reported for TCGA, and we observed merely 37.2% in our cohort. However, similar frequencies were observed for KRAS with 40.8% (compared to 39.8%) [40]. Compared to mutation frequencies reported by Lee et al. for *APC* (60%) and *KRAS* (49%), no apparent differences to our results were seen [41]. Similarly, the mutation frequency published by Burgenske et al. for *PIK3CA*, 15–25%, did not differ from our observation (17.7% in the HROC-Xenobank) [42].

When focusing on hypermutated tumors, the mutation frequency in TCGA increased to 57.5% for *BRAF*. The overrepresentation of hypermutated tumors in our cohort most likely explains the higher frequency of *BRAF* mutations observed in our cohort.

Mutation patterns of CRC from adolescent and young adults are also different. Tricoli et al. stated that in tumors of younger patients, genes were mutated significantly more frequently, particularly genes associated with DNA repair pathways BRCA2 (39% vs. 3%) and RAD9B (22% vs. 0%), as well as the cell-cycle checkpoint kinases ATM (35% vs. 7%) and ATR (48% vs. 13%). Despite the limited number of mutations associated with DNA repair pathway genes in our cohort, some of the HROC-Xenobank models might yet be interesting for functional analyses in that particular area of research.

In addition, many of our PDX models carry a higher number of mutations with uncertain significance than the original tumors, whereas Abdirahman et al. did not observe such additional mutations in their serially transplanted tumors [32]. Brown et al. suggested that mutations detected in PDX but not in original patient tumor samples could reflect hard to detect low-frequency clones in the original tumors [43].

The data from the German cancer registry ZfKD (Zentrum für Krebsregisterdaten) pointed out that 56% of patients with advanced CRC were male. The mean age among male patients ranged from 67.6 to 68.3 years and among female patients from 70.6 to 71.0 years [44]. The mean age of our study population is 69.7 (range 30 to 98) and thus lies precisely within the ZfKD ranges. Moreover, the male percentage of our study population is, with 56.8% of patients being male, the same as reported by the ZfKD. Tumor UICC staging was 11% (stage I), 29% (stage II), 27% (stage III), and 33% (stage IV) in our study, compared to 19.5% (stage I), 29% (stage II), 30% (stage III), and 21.5% (stage IV) in the general German CRC population [45]. Thus, the proportion of CRC stage II exactly matches, and the percentage of stage III nearly matches, the general German CRC population. However, CRC stage IV was considerably overrepresented and stage I considerably underrepresented in our study, as compared to the normal distribution of CRC stages in Germany. Partly, this is simply attributable to the fact, that our biobank collection is restricted to cases with sufficient tumor material available upon diagnosis in the pathology; therefore smaller, lower-staged cases frequently must be excluded. Additionally, all cases were collected at a university center, which is typically also biased towards more advanced, higher staged cases by the referring doctors.

The molecular subtype analyses revealed that our HROC-Xenobank cohort represents the common CRC subtypes [28]. However, spMSI-H and LS cases are overrepresented. This can best be explained by the fact that MSI tumors engraft significantly better than MSS tumors. Such a discrepancy in engraftment rates linked to the MS status was also reported for gastric cancer with 55.93% vs. 23.64%; *p* < 0.0001 [29]. This improved biological fitness in the xenograft environment was also highlighted by the fact that these PDX grew out faster, and fewer mice were necessary to generate ample amounts of backups. Our group previously described the molecular subtype (*p* = 0.003), especially the MS status (*p* = 0.001), as a potent parameter likely to influence the success rate of PDX establishment from CRC resection specimens [8]. It is also of relevance, albeit to a lesser extent, that MSI-H PDX tumors were not often highly necrotic and were, thus, rarely excluded from our final cohort due to this undesirable characteristic.

We want to emphasize the fact that the HROC-Xenobank has, already in its establishment phase, supported “standard” in vivo studies [46,47], detailed molecular pathway investigations [48], biomarker studies, and more basic studies which took advantage of snap-frozen samples [49]. A first pre-clinical PDX trial has been started in-house, preceded by a dose finding study [34]. Due to the fact that all samples have been collected exclusively after informed consent of all patients and the irrevocable anonymization of any personal or clinical data, the use of the HROC-Xenobank models is not restricted by the General Data Protection Regulation in the EU.

## 5. Conclusions

In summary, this study succeeded in generating 125 individual PDX models with sufficient numbers of vital and snap frozen aliquots as well as FFPE material. The preservation of the original tumor’s biological features such as microarchitecture and pathomorphology as intrinsic features of the individual models remain stable for several passages. Moreover, we were able to confirm the high concordance of pathomorphological as well as mutation patterns of the HROC-Xenobank to the underlying clinical case series. This enables the selection of individual models according to desired features and allows future investigations, such as pre-clinical PDX trials and detailed molecular pathway investigations, with well-characterized samples. Notably, the models of the HROC-Xenobank are available upon reasonable request.

## Figures and Tables

**Figure 1 cancers-13-05882-f001:**
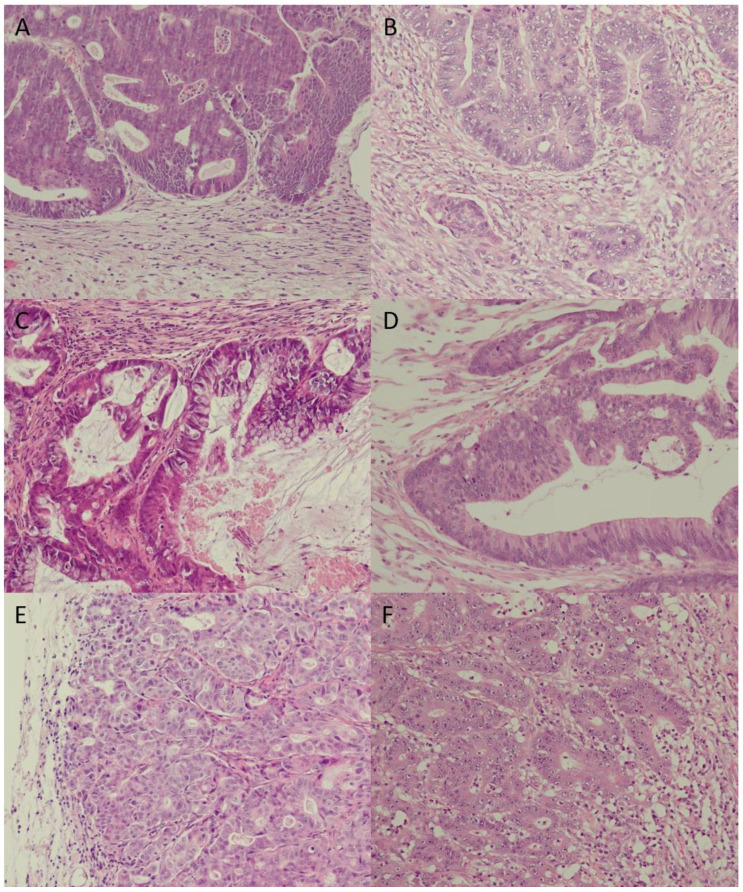
Comparison of primary tumor vs. PDX tumor in 20-fold magnification: (**A**) = HROC172 primary tumor, (**B**) = HROC172 T2 M2; (**C**) = HROC260 primary tumor, (**D**) = HROC260 T2 M5; (**E**) = HROC386Tu1 primary tumor, (**F**) = HROC386Tu1 T1 M1. In the case of HROC172, PDX cytomorphology and architecture match the primary tumor—stroma desmoplasia and tumor budding were markedly reduced; in the case of HROC260, PDX cytomorphology and architecture match the primary tumor—villous-mucinous structure was also reproduced; and, in the case of PDX HROC386Tu1, cytomorphology and architecture of the primary tumor was reproduced precisely.

**Figure 2 cancers-13-05882-f002:**
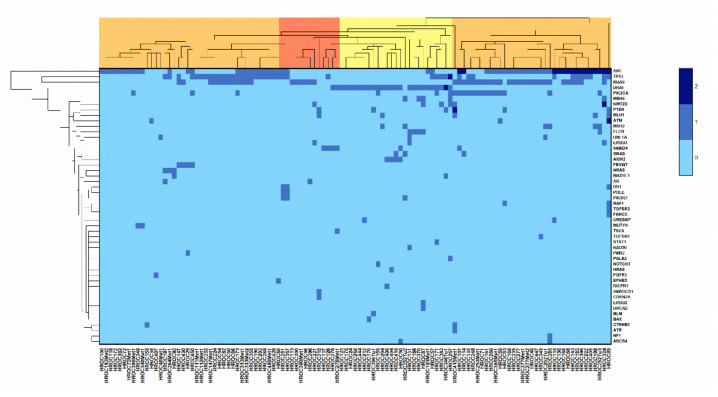
Unsupervised cluster analysis for all investigated tumors concerning the pathogenic or likely pathogenic mutations, with the following parameters: cluster method = Furthest Neighbor; distance type = Euclidean.

**Figure 3 cancers-13-05882-f003:**
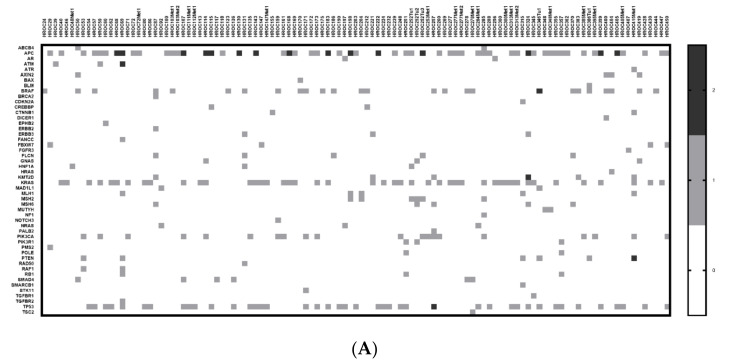
(**A**) = heat map. (**B**) = mutation frequency; (**A**) illustrates the number of pathogenic and likely pathogenic mutations per gene and case in a heat map, and the mutation frequency for each gene calculated out of this data is illustrated in (**B**).

**Table 1 cancers-13-05882-t001:** Sequences of STR primers.

Primer	Sequence
D5S818 for	5′-HEX-GGT GAT TTT CCT CTT TGG TAT CC-3′
D5S818 rev	5′-AGC CAC AGT TTA CAA CAT TTG TAT CT-3′
D7S820 for	5′-HEX-ATG TTG GTC AGG CTG ACT ATG-3′
D7S820 rev	5′-GAT TCC ACA TTT ATC CTC ATT GAC-3′
D16S539 for	5′-HEX-GGG GGT CTA AGA GCT TGT AAA AAG-3′
D16S539 rev	5′-GTT TGT GTG TGC ATC TGT AAG CAT GTA TC-3′
D13S317 for	5′-HEX-ATT ACA GAA GTC TGG GAT GTG GAG GA-3′
D13S317 rev	5′-GGC AGC CCA AAA AGA CAG A-3′
vWA for	5′-6-FAM-GCC CTA GTG GAT GAT AAG AAT AAT CAG TAT GTG-3′
vWA rev	5′-GGA CAG ATG ATA AAT ACA TAG GAT GGA TGG-3′
TPOX for	5′-6-FAM-ACT GGC ACA GAA CAG GCA CTT AGG-3′
TPOX rev	5′-GGA GGA ACT GGG AAC CAC ACA GGT TA-3′
THO1 for	5′-6-FAM-ATT CAA AGG GTA TCT GGG CTC TGG-3′
THO1 rev	5′-GTG GGC TGA AAA GCT CCC GAT TAT-3‘
CSF1PO for	5′-6-FAM-AAC CTG AGT CTG CCA AGG ACT AGC-3′
CSF1PO rev	5′-TTC CAC ACA CCA CTG GCC ATC TTC-3′
Amelogenin for	5′-ACC TCA TCC TGG GCA CCC TGG TT-3′
Amelogenin rev	5′-TAMRA-AGG CTT GAG GCC AAC CAT CAG-3′

**Table 2 cancers-13-05882-t002:** Molecular subclasses of the 125 investigated PDX, listed with total amount and percentage.

Molecular Subclass Determination (*n =* 125):		
CIN	65	52%
spMSI-H	29	23.2%
CIMP-H, non MSI	10	8%
CIMP-L, non MSI	10	8%
Lynch syndrome	10	8%
Neuroendocrine tumor	1	0.8%

## Data Availability

The data and materials are available from the corresponding author upon reasonable request.

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
