# Peer review of "The HROC-Xenobank—A High Quality Assured PDX Biobank of >100 Individual Colorectal Cancer Models"

_cancers, 2021, doi:10.3390/cancers13235882_

Round 1

Reviewer 1 Report

Well written and interesting paper.

Author Response

We want to thank this reviewer for this exceptional encouraging assessment.

Reviewer 2 Report

Matschos et al. describe the efforts to establish a comprehensive and quality assured tissue databank consisting of colorectal cancer (CRC) tissue that has been implanted subcutaneously in mice and passaged further up to 10 times. A number of clinical and molecular data are stored from each patient sample, including histopathology and DNA sequencing. This truly requires a massive organizational well conducted work to keep this biobank going. There is no doubt that the stored tissue will be valuable for research studies on the treatment of CRC. Importantly, the databank tissues are available for cooperative research approaches.

Major point

One major point concerns the tissue for collaboration studies. I am certain that several researchers around the world would be interested in such collaborations. Personal data will be connected with the tissue that researchers may receive from the databank. It should be discussed in more detail how this can be performed within the General Data Protection Regulation (GDPR) in the EU. Is the tissue available to researchers from the whole world, or is it only within the EU or within Germany? Which legal documents are required if it outside of Germany (or inside)? In some European countries the regulation (GDPR) is interpreted very strictly and handling of personal data requires legal documents in every single step when data is not irrevocably anonymised.

Minor point

A minor point is that the definition of HROC in the name HROC-Xenobank is first given on p. 6, paragraph 6.1. The definition should be given when the abbreviation appears first time. The title should also be changed since the abbreviation is used in the title.

Author Response

First, we want to thank this reviewer for not only a very encouraging assessment but also for the hint towards potential problems arising from the novel European GDPR. We modified the last part od the discussion accordingly. It now reads: "Due to the fact that all samples have been collected exclusively after informed consent of all patients and the irrevocable anonymization of any personal or clinical data, the use of the HROC-Xenobank models is not restricted by the General Data Protection Regulation in the EU.". 

We want to comment further our legal department did not see any problems when the two facts are given: informed consent and anonymization of data given out to any third party. Thus, also cooperations with partners outside of the EU are not problematic.

The second comment of this reviewer concerning the late introduction of the HROC definition was also helpful - we modified the manuscript and introduced the definition in the Introduction part. 

However, we still would like to stick to the chosen title which uses also the abbreviation HROC.

Reviewer 3 Report

This is a well written descriptive manuscript describing a new large set of complementary models. Figure 1 has largely been utilized in a JOVE publication (DOI: 10.3791/62065) by the same group.

Author Response

We first want to thank this reviewer for this exceptional encouraging assessment.

Concerning the comment on Figure 1, we want to respond that of course, both figures (Figure 1 of the present manuscript and the Figure in the JoVE publication) describe a very similar procedure and have been generated using the very same graphic programs. But they are still very different concerning the details etc.

Round 2

Reviewer 2 Report

The authors have now clearly stated that the tissue is not restricted by GDPR in EU and is available for researcher worldwide. This is absolutely great.

A minor point is the HROC abbreviation in the title and text which is not obvious for the general reader. I understand that the authors would like to keep it in the title, so my compromise is to define the abbreviation in the abstract. This would clarify things soon for the interested reader. Also, CRC in the title should be written as colorectal cancer.

Author Response

All points raised from reviewer #2 have now been fully addressed.

We wamt to thank this reviewer again for his comment concerning the GDPR in the EU. It really might push the subsequent use of the HROC-Xenobank all around the world.